# Expansion of MyDispense: A Descriptive Report of Simulation Activities and Assessment in a Certified Pharmacy Technician Training Program

**DOI:** 10.3390/pharmacy11010038

**Published:** 2023-02-16

**Authors:** Cassandra R. Doyno, Lisa M. Holle, Renee Puente, Sharee Parker, Lauren M. Caldas, Barbara Exum

**Affiliations:** 1Department of Pharmacy Practice, University of Connecticut, Storrs, CT 06269, USA; 2Department of Pharmacy Services, Yale New Haven Hospital, New Haven, CT 06510, USA; 3Department of Pharmacotherapy and Outcomes Science, Virginia Commonwealth University, Richmond, VA 23298, USA; 4Department of Pharmaceutics, Virginia Commonwealth University, Richmond, VA 23298, USA

**Keywords:** simulation, MyDispense, certified pharmacy technician training, PTAC, Pharmacy Technician Accreditation Commission, assessment

## Abstract

Background: Yale New Haven Health (YNHH) implemented a pharmacy technician training program in 2016. The curriculum includes 14 weeks of combined didactic and simulation hours (280 h in total), followed by 360 h of experiential learning. MyDispense, an online pharmacy simulation, allows students to develop and practice their dispensing skills in a safe environment with minimal consequences for mistakes. We describe a novel innovation, expanding the functionality of MyDispense to the training of pharmacy technicians. Methods: Technician training coordinator, supervisor, faculty members with experience in MyDispense, and experiential pharmacy students created cases within the MyDispense software that were targeted towards pharmacy technician activities. Activities were aligned with current American Society of Health-System Pharmacists (ASHP)-Accreditation Council for Pharmacy Education (ACPE) Accreditation Standards for pharmacy technician education and training programs. Results: A total of 14 cases were developed to be utilized in student technician training, and account for approximately 14 h of simulation. Conclusions: MyDispense is an innovative software that could allow students to access and complete exercises, and to continue developing dispensing skills in a safe, remote environment. We identified similarities between activities performed by student pharmacists and student pharmacy technicians, expanding MyDispense to a new learner group to practice, develop and be assessed on dispensing skills within their scope, as part of a formal technician training program and in preparation for the Pharmacy Technician Certification Examination (PTCE).

## 1. Introduction

In the American Society of Health-System Pharmacists (ASHP)’s Practice Advancement Initiative 2030 (PAI), multiple recommendations have been made for pharmacy technician education and training that will assist in advancing pharmacy practice in health systems [1,2]. A collaboration between the ASHP and the Accreditation Council for Pharmacy Education (ACPE) was responsible for the development of Accreditation Standards for pharmacy technician education and training programs [3,4]. By publishing standards, and with the formation the Pharmacy Technician Accreditation Commission (PTAC), the aim is to ensure the development of a competent workforce of pharmacy technicians by providing entry and advanced-level training programs in addition to providing specific criteria to aide in development of new training programs and promote improvement in established programs [3]. This will create a standard of expectation for training programs that previously was lacking [5,6,7]. As a result of such efforts, technician programs around the country have focused on increased curriculum offerings (didactic, simulation, and experiential training) to ensure pharmacy technicians who enter the workforce are ready to fulfill their duties within the healthcare team [8,9,10].

The ASHP/ACPE Accreditation Standards curriculum requirements are outlined in three main educational components: the didactic, simulation, and experiential. Each component has a minimum number of hours whether developing an entry or advanced-level program, with flexibility to add additional hours to certain areas [4]. Simulation components provide hands-on, realistic experiences to complement the classroom-based, didactic learning. Institutions have developed strategies for both asynchronous simulation experiences, such the act of calling patients and fact-finding within package inserts, where students record themselves completing these tasks, and synchronous onsite simulations, including reconstitution of oral or topical products [9]. However, within outpatient and retail-based learning experiences, barriers still exist within the realm of simulation, as students in training are not able to experience key aspects of processing and dispensing a prescription. These activities are generally observed only during experiential learning, given legal constraints, and require familiarity with electronic prescription processing systems. This creates a learning environment in which pharmacy technician students could potentially make mistakes that impact patients and pharmacy teams. This increases the burden on the pharmacy team and stress on the students.

MyDispense, an online pharmacy simulation platform, was developed in 2010 by faculty of Pharmacy and Pharmaceutical Sciences at Monash University in Australia for the purposes of pharmacy student education. This platform allows students to develop, practice and be assessed on their dispensing skills in a learning environment without real-world consequences of mistakes [11]. MyDispense, utilized by approximately a third of the United States Schools of Pharmacy [12], consists of customizable modules that faculties at schools of pharmacy create with the intent for students to practice tasks ranging from novice to advanced skill levels, reinforcing concepts that are key to the medication use process such as prescription processing and patient safety. Each exercise enables the student to attempt a task, receive feedback and retry the exercise to correct any mistakes. Simulation that is possible with this unique program has an added advantage in that it can be completed asynchronously or synchronously, or using a hybrid approach [13]. Additionally, MyDispense can be utilized as a form of assessment. Currently, MyDispense is utilized in 186 schools of pharmacy across 34 countries, helping over 25,000 pharmacy students worldwide [13]. MyDispense has been received well by pharmacy students and seen as positive prior preparation for experiential training [14,15,16].

In addition to the customizability of MyDispense, benefits can be seen for both the instructors and students. For instructors, these simulations update automatically, without instructors needing to manually update or set up the materials for the simulation. Additionally, for both faculty and students, MyDispense cases allow for remote learning via simulation (which is key in a post-COVID era), which allows flexibility of learning environment to fit the health and life needs of the faculty and students. MyDispense also alleviates the pharmacy team from having pharmacy technicians’ first dispensing opportunities occur in the pharmacy wherein patient care may potentially be impacted. While MyDispense was designed for pharmacy students, its adaptability and customization to an independent program’s needs allows the potential for it to expand to other teaching and learning environments with similar training objectives, as has been seen in nursing programs [17] and medical schools [18]. This functionality was recognized by faculty members from the schools of pharmacy at the University of Connecticut (UConn) and Virginia Commonwealth University (VCU), in collaboration with Yale New Haven Hospital (YNHH) Pharmacy Technician Training Program. We describe a novel innovation, expanding the functionality of MyDispense to the training of student pharmacy technicians.

## 2. Materials and Methods

The Pharmacy Technician Training Program at YNHH was established in 2016, as a product of blending ASHP/ACPE national standards with the needs of their hospital system [8,19]. The 23-week program curriculum combines didactic and simulation hours (280 h in total), followed by 360 h of experiential learning [8]. The idea of bringing MyDispense to a pharmacy technician training program was initially brought about by a UConn student on experiential rotations, who had a history of MyDispense use, as it was utilized within their pharmacy curriculum. Given their knowledge of the pharmacy technician training program at YNHH and the similarity of learning objectives with regard to concepts in community/retail settings, they inquired about the possibility of adapting MyDispense exercises to enhance pharmacy technician student learning. Given the collaborative nature of pharmacy schools who utilize an online MyDispense case repository, exercises utilized in PharmD curricular courses could easily be modified to reflect how a pharmacy technician would experience similar scenarios.

Our project occurred over three phases. During phase one, faculty members from UConn further brainstormed the idea during a research collaboration session at the MyDispense Global Symposium, in collaboration with faculty from VCU, who are implementing a new technician training program to meet the need for accredited pharmacy technician programs in Virginia. Starting in July 2022, all pharmacy technicians in Virginia must receive their training from a program accredited by ASHP and ACPE [20]. The VCU faculty felt this was an opportunity to assist in training the next generation of pharmacy technicians, and schools of pharmacy are well positioned to offer training to pharmacy technician training programs [21,22]. This growing career path increases at an average of 5% each year [23], and the school is prepared to offer advanced training that includes all three required portions of the accredited curriculum. The VCU faculty discussed with UConn the ability to expand their use of MyDispense to this new program as a part of their simulation activities.

Once contact was established with leadership within the program at YNHH, phase two began with forming a development group, which consisted of a pharmacy technician training coordinator (1) and supervisor (1), faculty members from UConn (2) and VCU (2), experiential pharmacy students (2) and a research pharmacy student (1) from UConn. A plan to create or modify pre-existing MyDispense cases to meet ASHP/ACPE standards and program objectives was established. Development meetings were held weekly, starting in September of 2022 for four weeks, then monthly thereafter until the anticipated implementation into the program’s simulation curriculum in January 2023.

Pharmacy students completing their Health Systems experiential rotation worked closely with coordinators to review ASHP/ACPE Outpatient Pharmacy Learning Objectives that are integrated into the simulation portion of the program curriculum in phase three of the project. Objectives that would be addressed utilizing MyDispense activities were identified. A blueprint was created (Table 1) to highlight the standards that MyDispense cases could be modified and/or built to address. An online MyDispense repository was reviewed to identify and import cases that were originally created for pharmacy student education, and instructions were provided on how to customize each case to technician training specifications (Table 1). Standards required a new build if it could not be directly mapped to a pre-existing case (Table 2). All case building and modification was completed through the MyDispense software. The School of Pharmacy faculty was available to assist in case development, as they were familiar with certain MyDispense administrative functions from their experience in pharmacy education courses. Lastly, phase four was the pilot of simulation cases and an assessment, completed by a cohort of six students in the YNHH technician training program. 

## 3. Results

A total of 14 cases were developed in MyDispense to be utilized in student technician training. Twelve standards/objectives were met by utilizing the developed cases, and account for 14 h of simulation within the program’s curriculum. Cases that were modified from the MyDispense online repository [24] originated from cases developed by schools of pharmacy from The University of Kentucky, University of Michigan, and University of Connecticut. Given the collaborative nature of MyDispense, the online repository is available to upload previously developed cases for purposes of sharing via the MyDispense website [24]. Participating universities have the ability import cases from the repository to their own platforms to use and/or modify. We were able to utilize cases shared in the repository to address all but four learning outcomes. Cases stemmed from various dispensing exercises encompassing scenarios that technicians in a community or retail setting are likely to encounter, such as identifying allergies and other health information, updating patient profiles, validating prescriptions to ensure all required information is included, knowing when to escalate questions to a pharmacist, label generation, ancillary labeling, recognizing controlled substance regulations, recalls, and considerations in insurance payments. Many of these scenarios otherwise would not be encountered as a student in training, or in training of an observation-only nature.

This project offered a unique opportunity for pharmacy student involvement, and incorporated two fourth-year pharmacy students completing experiential rotations and a second-year pharmacy research student. The total combined time spent by the three pharmacy students developing the blueprint and carrying out MyDispense case building and modification was around 112 h. The collaborative meetings between faculty, pharmacy technician training program staff and students accounted for 4.5 h of the workload.

As the basis of this paper was to describe an innovative use within the MyDispense platform, our group has many goals for future directions. Our first pilot was done with a cohort of six students, and included simulation exercises alongside an assessment portion which incorporated many learning objectives. At time of this publication, simulations were completed over 8 total hours allotted within the course. Feedback sessions are planned and will allow for us to adjust any exercises. We plan to collect data to further describe the use of simulation exercises and assessments within the technician training program on a larger sample size. With that, we aim to analyze and assess achievement of learning objectives and first-pass success of board examinations after multiple cohorts of the technician training program have utilized this MyDispense functionality.

## 4. Discussion

Although simulation is a required component of the ASHP/ACPE Accreditation Standards curriculum’s educational requirements, published reports of the simulation types employed in pharmacy technician training programs are limited to inpatient settings or customer interaction in the community pharmacy setting [8,9]. This is the first report, to our knowledge, of using a community pharmacy-based simulation program that focuses on the role of the technician in a community pharmacy setting. With over 43,500 openings for pharmacy technicians projected each year by the U.S. Bureau of Labor Statistics, and over 3000 new certified pharmacy technicians (CPhTs) each year, this innovative simulation could be quite useful for training pharmacy technicians who seek community pharmacy technician experience and/or positions [25,26].

The functionality of MyDispense, with its ability to have the learner communicate with patient or providers and identify when a pharmacist needs to become involved, collect patient information, process prescriptions (including assessing prescriptions for completeness, entering prescription information into the electronic pharmacy system, selecting the product and labeling the product), and prioritize tasks, is aligned with many of the ASHP/ACPE Model Curriculum for Pharmacy Technician Education and Training Programs’ outpatient pharmacy learning objectives. Furthermore, MyDispense offers a platform to eliminate some of the gaps in simulation opportunities that currently exist within pharmacy technician training programs, such as (1) processing and handling of medications and medication orders (2) procurement, billing, reimbursement, and inventory management, and (3) patient and medication safety.

We did encounter some limitations to our project. At this time, simulation activities are limited in scope to the outpatient community setting; therefore, technician roles that are prominent in the inpatient hospital setting are unable be simulated at this time. However, an inpatient version of MyDispense is in development. Additionally, the scope was limited within certain outpatient scenarios, as students in training are unable to practice prescription processing through various insurance or billing scenarios because this is not a feature available via MyDispense. While the repository does allow for sharing of cases nationwide by participating colleges of pharmacy, we still needed to develop four cases to suit the simulation needs of the program. As disclosed in the results section, case modification and development can take a significant amount of time, which was completed by fourth-year pharmacy students as part of their experiential rotations. However, administrative tasks were completed by a faculty member and training program coordinator, which did require some workload commitments and should be considered when deciding to implement such programs. Finally, navigation of institutional cybersecurity and information technology support service requirements is needed when implementing MyDispense, which does require an initial investment of time [13]. While historically, this has not been a barrier in academic settings, who to our knowledge have successfully implemented MyDispense without issues, it may be a barrier to some health systems. Therefore, institutional information technology services should be consulted to discuss compliance with institutional policies when using MyDispense for technician training programs.

A recent survey of 204 pharmacy technicians showed that over 60% of pharmacy technicians prefer to learn using kinesthetic learning methods (i.e., learning by doing) [27]. MyDispense is perfectly suited as a kinesthetic learning method in that it allows the learner to learn while doing. In MyDispense, learners are able to gather information from patients, input prescriptions, and even simulate labeling medication bottles. This system also allows the learner to practice and then receive immediate feedback on their performance. If desired, they can reset the exercise and complete again (as many times as they choose). Additionally, in this survey, respondents reported using a review book and on-the-job training as the most frequent methods of studying for the pharmacy technician certification exam (PTCE). MyDispense is also well suited for PTCE preparation as it can allow the learner to practice at their own pace (because it can be used asynchronously), and the immediate, targeted feedback has been shown to improve learners’ rates of learning [13,28].

Another benefit of using MyDispense in a technician training program is a reduction in faculty workload. Typically, the medication dispensing process is taught in-person or during an experiential rotation, thus requiring the faculty and learner to be onsite within a classroom or pharmacy. However, MyDispense can allow the student to practice many of these skills asynchronously [29]. The faculty does not need to be present since the feedback is given automatically to the learner, and the learner can repeat the exercise to correct mistakes, further enhancing learning. Similarly, MyDispense can be used for assessments, which can be limit the time spent on grading since this is also automated. Additionally, MyDispense decreases the burden of purchasing and the faculty classroom set-up time needed for in-person dispensing; all the medication images are on the program, and can be updated to the necessary products. Faculty do not need to print or set up dispensing simulations that might need to then be taken down and stored until the next offering. With MyDispense, there is a significant decrease in the scheduling and administration responsibilities of faculty for each offering of these classroom activities. The freely available repository of exercises can also decrease faculty workload. From the repository, exercises can be imported for use or imported and edited for use. This was the approach taken during this project to create technician-specific exercises; existing exercises were imported and edited to be appropriate for a technician learner. While MyDispense can be used asynchronously, it could also be used in a hybrid format to increase the feasibility of meeting certain objectives, specifically to meet ASHP/ACPE simulation and feedback requirements for learning. For example, MyDispense does not have the ability to compound non-sterile medications, but it could be used in conjunction with a hands-on laboratory exercise, wherein MyDispense is used to prepare medication labels and a hands-on laboratory exercise is used for the learners’ preparation of non-sterile medication [30].

## 5. Conclusions

In conclusion, MyDispense has the ability to fill a gap that currently exists in many pharmacy technician training programs: simulation of processing and dispensing a prescription in a community pharmacy setting. MyDispense is an innovative software that could additionally allow students to access exercises and continue to develop dispensing skills in a safe, remote environment. It can be not only useful for completion of a formal technician training program either synchronously or asynchronously, but also in preparation for the PTCE. It can contribute to reduced faculty workload and provide additional educational learning opportunities that in other settings have been shown to improve educational outcomes. As our program is implemented, we will seek to evaluate learning outcomes with the use of MyDispense for technician training, expand exercises to achieve all learning objectives related to community pharmacy, and allow for technician learners to practice these prior to taking the PTCE exam.

## Figures and Tables

**Table 1 pharmacy-11-00038-t001:** Blueprint for technician training MyDispense exercises [3,24].

ASHP/ACPE Standards [3]	MyDispense Activity/Patient Case Description (Case School of Origin)	Modifications to Make
3.1: Assist pharmacists in collecting, organizing, and recording demographic and clinical information for the pharmacists’ patient care process.	Dispensing exercise (University of Kentucky): This exercise assesses the student’s ability to do the following: (1) Dispense appropriate medication (2) Identify patient allergy to prescribed drug (3) Contact prescriber for appropriate alternative antibiotic.	(1) Change: Take out contacting the provider for alternative agent → not the technician’s role (2) Keep: Allergy identification/update that would warrant that prescription inappropriate to hand over (3) Add: Letting the pharmacist know of the patient’s allergy and not dispensing the medication, asking patient for other updated demographic information (PHI, new medical conditions, new address, etc.) (4) What we want the student to get out of this exercise: identifying allergies, asking PHI, updating profiles to most current information for each patient
3.2: Receive, process, and prepare prescriptions/medication orders for completeness, accuracy, and authenticity to ensure safety.	Dispensing exercise (University of Kentucky): This exercise assesses the student’s ability to do the following: (1) Dispense appropriate medication (2) Identify and fix a dosing error that would otherwise result in a toxic overdose.	(1) Change: Take out fixing a dosing error that would otherwise result in a toxic overdose (2) Keep: dispensing appropriate medication (picking, counting, labeling) (3) Add: instead of fixing a dosing error, maybe have students follow up with a provider regarding a missing quantity or strength of medication before dispensing (4) What we want the students to get out of this exercise: Prescription processing
	Dispensing exercise (University of Kentucky): This exercise assesses the student’s ability to do the following: (1) Dispense appropriate medication (2) Identify and fix a prescription error (date missing and request for fill prior to vacation; contact physician authorization).	(1) Change: none (2) Keep: everything should stay the same (3) Add: Check fill history to make sure patient is not filling too soon, ask about counseling for new medication (4) What we want the students to get out of the exercise: Looking at fill history, patient data, identifying that the date is missing and script cannot be filled without clarification
3.3: Assist pharmacists in the identification of patients who desire/require counseling to optimize the use of medications, equipment, and devices.	Dispensing exercise (University of Michigan): This exercise assesses the student’s ability to do the following: (1) Dispense appropriate medication (patient comes to refill warfarin) (2) Identify patient asking question (taking particular OTC) that requires pharmacist to answer	(1) Change: technician does not provide the counseling (patient asks “ can I take over the counter herbal product St. John’s wort because Dr. Oz says it is good for depression?” (2) Keep: the two medications in question and the question being asked about them (3) Add: need an option to hand over the patient question to the pharmacist (4) What we want the students to get out of this exercise: knowing when to hand over a patient question to the pharmacist
3.4: Prepare patient specific medications for distribution.	Dispensing exercise (University of Connecticut): This question assesses the student’s ability to do the following: (1) Calculate the appropriate dose of antibiotic for the patient (2) Dispense the correct medicine to the patient (3) Chose the appropriate ancillary labels for the medicine to be dispensed (4) Answer patient queries regarding dosing of pediatric acetaminophen	(1) Change: Do not need the patient queries regarding dosing of pediatric acetaminophen (2) Keep: Dose calculation, dispensing, ancillary labeling (3) Add: Allergy check (4) What we want the students to get out of this exercise: Calculating a dose, dispensing process, label generation, ancillary labeling, non-sterile compounding of pediatric liquid antibiotic
	Dispensing exercise (University of Michigan): This question assesses the student’s ability to do the following: (1) Dispense appropriate medication (simple dispensing exercise with no issues. Montelukast)	(1) Change: none (2) Keep: everything should stay the same (3) Add: new medication, ask for counseling (4) What we want the students to get out of this exercise: dispensing process and label generation
3.7: Assist pharmacists in the monitoring of medication therapy.	Dispensing exercise (University of Michigan): This question assesses the student’s ability to do the following: (1) Identify that patient is overdue for medication refill of a medication when he presents for refill of another medication (2) Know when to get pharmacist involved	(1) Change: none (2) Keep: Metformin is overdue; even though he is presenting for another medication, patient is not filling medications according to how often the directions would indicate he should be. Tech is to identify that patient’s refill is late and the directions would have him refilling every month (3) Add: identify the need for pharmacist input on medication recommendation aspect (4) What we want the students to get out of this: addressing adherence, being able to look into the patient fill profile to monitor their adherence based on fill history, getting the pharmacist involved on the recommendation question
3.13: Use current technology to ensure the safety and accuracy of medication dispensing.	This standard is met in all dispensing activities in the MyDispense software - Students are selecting medications from the shelves and scanning them to verify it matches with the prescription at hand. Students have access to databases in the MyDispense platform	
3.17: Assist pharmacists in preparing medications requiring compounding of non-sterile products	Dispensing exercise (University of Michigan): This question assesses the student’s ability to do the following: (1) Calculate the appropriate dose of antibiotic for the patient (2) Dispense the correct medicine to the patient (3) Choose the appropriate ancillary labels for the medicine to be dispensed (4) Answer patient queries regarding dosing of pediatric acetaminophen	(1) Change: Students do not need the patient queries regarding dosing of pediatric acetaminophen (2) Keep: dose calculation, dispensing and compounding liquid antibiotic, ancillary labeling (3) Add: allergies (4) What we want the students to get out of this exercise: calculating a dose, dispensing process, label generation, ancillary labeling, non-sterile compounding of pediatric liquid antibiotic
3.19: Explain accepted procedures in inventory control of medications, equipment, and devices.	Dispensing exercise (University of Michigan): This exercise assesses the student’s ability to do the following: (1) Identify medication on shelf is expired (2) Contact patients to inform them when medication will be available.	(1) Change: take out dosing error on this exercise (2) Keep: the patient needs the antibiotic ASAP (3) Add: when the tech goes to the shelf to retrieve medication, the medication on hand is expired; the technician has to inform patient that it will be on order for tomorrow (4) What we want the students to get out of this exercise: being able to read sig codes, and translating instructions to ensure they make sense
5.1: Describe and apply state and federal laws pertaining to processing, handling and dispensing of medications including controlled substances.	Dispensing exercise (University of Connecticut): This exercise assesses the student’s ability to do the following: (1) Dispense the correct C-IV medication.	(1) Change: none (2) Keep: the same (3) Add: nothing to add (4) What we want the students to get out of this exercise: processing, handling and dispensing of medications including controlled substances, knowing the rules and regulations surrounding controlled substance prescriptions

ADR, adverse drug reaction; PHI, protected health information.

**Table 2 pharmacy-11-00038-t002:** Simulations to create in MyDispense [3,24].

ASHP/ACPE Standards [3]	Simulation to Create in MyDispense
3.6: Assist pharmacists in preparing, storing, and distributing medication products including those requiring special handling and documentation	Point of the simulation: drugs with mandated REMS Terms/topics that we are looking to touch on: REMS programs, why a REMS program is indicated and who is involved How to incorporate: Patient is presenting with a prescription for Accutane. The technician should be able to identify that the prescription requires REMS processing and should interact with the patient via the question-asking section to explain this requirement and how the process works. Patient can ask these questions and technician can type in their answers accordingly - What is REMS and what is the point of having it in place for Accutane - Who is involved in the REMS - What does it require and how is the patient involved in it
3.12: Explain procedures and communication channels to use in the event of a product recall or shortage, a medication error, or identification of another problem.	Point of the simulation: simulate a product recall and the technician’s role in using procedures and proper communication in the event that one occurs Terms/topics that we are looking to touch on: Recall, communication, contact tracing How to incorporate: case should involve a drug recall for a medication that is stocked in the shelving area: - Technician should navigate to the shelf and “recall” the affected lot numbers - Technician should get a report on the computer that indicates which patients have been dispensed that medication and proceed with proper procedures (calling the patients and documenting) to ensure that all who are affected are contacted.
3.14: Collect payment for medications, pharmacy services, and devices.	Point of the simulation: cash register interaction and responsibilities pertaining to the technician Terms/topics that we are looking to touch on: deductible, copay, Medicare donut holes How to incorporate: Build the conversation in the patient question section - Have the patient ask the question of why the cost of the medication is much more than normal at the start of the new year. Technician explains that the patient is meeting their deductible which is an additional amount of money on top of their copay for the medication - A Medicare patient is wondering why a prescription that usually costs him/her $7.00 is $300.00 at pickup. Technician explains that the patient is in the donut hole or a coverage gap which indicates there is a temporary limit on what the prescription plan will cover. Not everyone reaches this coverage gap; however, for those that are affected, it begins once a certain amount of money has been paid by the patient and the plan.
4.2: Apply patient and medication safety practices in aspects of the pharmacy technician’s roles.	Point of the simulation: recognizing common adverse drug events and duplicate therapies pertaining to their top 200 drug list Terms/topics that we are looking to touch on: duplicate therapies, ADRs of commonly prescribed medications, MedWatch ADR reporting How to incorporate: Patient presents with a new script for a beta blocker. Patient fill history reflects that they are already on another beta blocker; therefore, they not indicated to start another agent within the class - Have the patient fill history consist of 10 drugs, one of which is a metoprolol, and have the patient present with a new script for timolol and look to fill it - Technician should be able to recognize that this would be a duplicate therapy Pregnant woman presents to pick up her husband’s methotrexate - Technician should be able to identify this medication as hazardous to a child-bearing woman and handle situation accordingly

ADR, adverse drug reaction; REMS, risk evaluation and mitigation strategies.

## Data Availability

Data sharing not applicable. No new data were created or analyzed in this study. Data sharing is not applicable to this article.

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
