# Peer review of "Expansion of MyDispense: A Descriptive Report of Simulation Activities and Assessment in a Certified Pharmacy Technician Training Program"

_pharmacy, 2023, doi:10.3390/pharmacy11010038_

Round 1

Reviewer 1 Report

In this article the authors describe the use of the MyDispense software in Certified Pharmacy Technician Training Program.

First, MyDispense was designed for pharmacy students but its adaptability and customization to a pharmacy technician training program is possible. This functionality was recognized by faculty members from schools of pharmacy at the University of Connecticut (UConn) and Virginia Common wealth University (VCU), in collaboration with Yale New Haven Hospital (YNHH) Pharmacy Technician Training Program.

The methodology used is well adapted. Students with coordinators worked closely  to review ASHP/ACPE Outpatient Pharmacy Learning Objectives . Objectives  addressed utilizing MyDispense activities were identified. A blueprint was created  to highlight the standards and an online MyDispense repository was reviewed to identify and import cases that were originally created for pharmacy student education, and instructions were provided on how to customize each case to technician training specifications

These are 14 cases that have been developed in MyDispense software to be used in the training of technician students. Twelve standards/objectives were achieved using the cases developed.

On the other hand, it should be noted that different pharmacy schools from the University of Kentucky, the University of Michigan and the University of Connecticut have participated in this program.

Author Response

Please see attachment. Responses to all reviewer comments are in red. 

Reviewer 2 Report

This article describes how Doyno and colleagues used the popular MyDispense community pharmacy simulation in a pharmacy technician training program. In their program, they modified existing simulation scripts intended for use by doctor of pharmacy students and created additional simulations with the result of producing MyDispense simulations appropriate for pharmacy technician trainees. They believe this is the first published evidence of using MyDispense or any other computer simulation in pharmacy technician training programs, and indeed a PubMed search does not show any obvious examples.

The article is well-written and describes the current scene for pharmacy technician training, especially given guidance from ASHP and ACPE, although the article seems more a case report or description of a program than original research. The 14 simulation scripts developed seem like appropriate additions to a pharmacy technician curriculum and address the objectives in the ASHP/ACPE document. Presumably these have been published and shared so that other programs may use them, making this article a significant contribution to the academic training of pharmacy technicians anywhere in the country.

While the article is well-written, there are some small changes and clarifications that could improve it:

1. The article should follow reporting guidelines appropriate to the intended article type. The CARE guidelines for reporting case reports may be a good starting point, but the SQUIRE-EDU or DoCTRINE guidelines may be more specific given the educational nature of the intervention. The article is already very close to the DoCTRINE checklist, but I would recommend a brief look.

2. Either the title or abstract should indicate that this is merely the description of an educational innovation, not a study.

3. The 4 numbered points in the abstract are awkward to read. These same points may be better suited as Background (or introduction), Methods, Results, Conclusions.

4. There should be no citations in the abstract. All 3 citations from the abstract are included in the main article, and these main article citations will suffice to credit the original ideas.

5. There are some minor lapses in grammar or word choice. For example, there should be a comma after (PAI) in the first sentence of the introduction. Also, "wrong doings" (page 2, line 74) may carry connotations of morally or legally unallowable actions. "Mistakes" may be a better choice. If there are additional language problems, I didn't notice them, but double-check me, please.

6. I found repeated references to hospitals and hospital systems distracting. MyDispense simulates a community pharmacy, as you have acknowledged later in the article, and while some skills are transferable to a hospital pharmacy setting, most of your simulation learning objectives apply most directly or exclusively to the community pharmacy setting. I was not clear on what would be the nature of your intervention until I arrived at your chart on page 4. Consider de-emphasizing that the intention of the program is to prepare students for hospital work, or else more explicitly describe why community pharmacy simulations are important in that preparation.

7. The sentence beginning on page 3, line 110 reading, "This statewide mission to guarantee the quality training for pharmacy technicians has improved the offerings and education of pharmacy technician trainees" is problematic. There is no citation to support your assertion. More, while Virginia's experience may be different, the general implications of assertion are hotly contested by many in the pharmacy profession. The counter-example shows a decline in the quality of students enrolling in (and thus graduating from) pharmacy technician programs - federal laws require programs to provide information on career outlook to prospective students, and pharmacy technicians have a very low salary with the same cost and duration of schooling compared to other health care technician jobs, driving many would-be pharmacy technicians to other career pathways.

8. The sentence beginning on page 3, line 114 reading, "Other programs have closed potentially due to not able to offer the expanded... requirements," is also problematic. Without a supporting citation and using the word "potentially" as a disclaimer, this is merely speculation. To the reader, it is too convenient and seems self-serving that you would speculate that other programs closed because they didn't have your intervention. Reports from ACPE that programs are scored Not Compliant on these standards or even anecdotal evidence that this caused other programs to close could bolster your case. Otherwise, you can attest to the difficulty of fulfilling these obligations without speculating about it causing other programs to close.

9. I understand how your methods sections is set up, but I still wish there was a way for you to explicitly state that your intervention is the modification of existing PharmD cases earlier in the methods. This would have clarified a lot for me as I read the rest of the methods section.

10. The results section is where I would love to see more attention to the DoCTRINE checklist. How many students have participated so far? How many tries did it take to clear each simulation on average? How long does each simulation take to complete? Has this improved over multiple offerings of the curriculum? Do you have any evidence of achievement of the learning outcomes from your tables? Is there any change in first pass rate in PTCB/ExCPT from before the simulation to after the simulation? Preceptor satisfaction for students completing experiential rotations? If you haven't offered the simulations in-class yet, this needs to be more explicitly stated. If you have, it would be nice to know as many details about the real use of the simulations as possible.

11. Are these openly published for other programs to use? Stating this and providing a link or directions for how to find these cases in the repository would be very powerful for the profession and your article's impact. I didn't notice a statement describing permission from Michigan or Kentucky to use their cases, much less to use them and keep the modifications proprietary, so please consider if it is ethical to do so.

12. The Discussion identifies the MyDispense activity as kinesthetic. While moving a mouse or pressing a keyboard are physical actions, this simulation would not be considered kinesthetic by psychologists (Game Studies. 2013;13(1)). Your next sentences seem to be to explain what makes this kinesthetic, but these attributes, repetition and immediate feedback, do not describe kinesthetic learning.

13. The discussion section does not identify any limitations to your intervention. Is there anything that potential adopters should consider? Do students initially resist the intervention? Are there still gaps in the open repository that need to be filled? When making a new case or modifying an existing one, does that take a significant amount of faculty time? Is the intervention poorly suited for simulating hospital settings?

14. The conclusion in the main article is great. The conclusion in the abstract states that the platform "[offers] a seamless transition to online learning..." and this was not demonstrated in the article.

Again, this was a well-constructed article, and an exciting intervention to describe - I was very interested in your work and want to learn more. My biggest complaints are that the article is missing some information that it probably should contain, and a few unsupported statements that could be discarded easily. Thank you for the opportunity to review your work!

Note, in case this explains any of the above comments: This reviewer is faculty at a PharmD program in the United States, and is familiar with and has worked with MyDispense although it is not used in the reviewer's own classes and lessons. The reviewer was previously a PTCB-certified technician, but did not attend a formal training program nor has the reviewer been involved with teaching at a formal training program for pharmacy technicians.

Author Response

(The authors gave the same response as above.)

Reviewer 3 Report

Interesting topic and nicely presented. Introduction is well written. Methods: I believe that this section needs to be more organized i.e. phases of the study. Results are clear and supported with tables which makes this section easy to follow. Discussion would be more informative if more studies were included and limitations of the study could be added

1. What is the main question addressed by the research? The authors provided a great example of using my dispense in teaching pharmacy technicians and illustrated that clearly so that the method they use is reproducible 2. Do you consider the topic original or relevant in the field? Does it address a specific gap in the field? Yes. It is specific. We use Mydispense in teaching our undergraduate pharmacy students and we are planning to have a bridging program for pharmacy technicians and I believe the mapping with ASHP/ACPE Standards is very helpful and could be used in creating similar cases 3. What does it add to the subject area compared with other published material? According to the literature that I looked at, this is the first time that my dispense is used for pharmacy technicians 4. What specific improvements should the authors consider regarding the methodology? What further controls should be considered? The methodology would be much organized if they write as phases. A suggestions for the authors to look at the American journal of pharmacy education as they have layout for reporting educational studies. 5. Are the conclusions consistent with the evidence and arguments presented and do they address the main question posed? Yes are supported by the study findings 6. Are the references appropriate? Yes, but they could more relevant studies, especially to the discussion section 7. Please include any additional comments on the tables and figures. All clear

Author Response

Interesting topic and nicely presented. Introduction is well written. Thank you so much for taking the time to review our work and providing this feedback.

Methods: I believe that this section needs to be more organized i.e. phases of the study.

Response: The methods section now has phases associated with different steps of project implementation.

Results are clear and supported with tables which makes this section easy to follow. Thank you very much!

Discussion would be more informative if more studies were included and limitations of the study could be added.

Response: Our literature search resulted in finding no published papers on this specific topic, therefore we feel this is an innovative project which will represent the first paper published on such implementation. Limitations were added to the discussion section (page 10, line 213-225).
